



# Using rainfall thresholds and ensemble precipitation forecasts to issue and improve urban inundation alerts

Tsun-Hua Yang[1], Gong-Do Hwang[1,2], Chin-Cheng Tsai[1], Jui-Yi Ho[1]

[1]Taiwan Typhoon and Flood Research Institute (TTFRI), National Applied Research Laboratories (NARLabs), Taipei, 10093, Taiwan

[2]Department of Atmospheric Science, National Taiwan University, Taipei, 10617, Taiwan

*Correspondence to*: Tsun-Hua Yang (tshyang@narlabs.org.tw)

**Abstract.** Urban inundation forecasting with extended lead times is useful in saving lives and property. This study proposes the integration of rainfall thresholds and ensemble precipitation forecasts to provide probabilistic urban inundation forecasts. Utilization of ensemble precipitation forecasts can extend forecast lead times to 72 h, preceding peak flows and allowing response agencies to take necessary preparatory measures. However, ensemble precipitation forecasting is time and resource intensive. Using rainfall thresholds to estimate urban areas' inundation risk can decrease this complexity and save computation time. This study evaluated the performance of this system using three hundred and fifty-two townships in Taiwan and seven typhoons during the period 2013-2015. The levels of forecast probability needed to issue inundation alerts were addressed because ensemble forecasts are probability based. This study applied six levels of forecast probability and evaluated their performance using five measures. The results showed that this forecasting system performed better before a typhoon made landfall. Geography had a strong impact at the start of the numerical weather modeling, resulting in the underestimation of rainfall forecasts. Regardless of this finding, the inundation forecast performance was highly contingent on the rainfall forecast skill. This study then tested a hybrid approach of on-site observations and rainfall forecasts to decrease the influence of numerical weather predictions and improve the forecast performance. The results of this combined system showed that forecasts with a 24-h lead time improved significantly. These findings and the hybrid approach can be applied to other hydrometeorological early warning systems to improve hazard-related forecasts.

## 1 Introduction

Flooding is one of the most destructive disasters in the world and results in enormous losses of life and property annually (Gruntfest and Handmer, 2001; Barredo, 2009; Hallegatte et al., 2013; Sampson et al., 2015). Global flood risk is likely to increase under climate change; as a result, numerous adaption strategies should be considered (Hirabayashi et al., 2013). Establishing an early flood warning system to reduce disaster losses is the most cost-effective solution of all of the structural and non-structural measures studied (Alfieri et al., 2012; Hallegatte, 2012).

Floods can be divided into four categorizes based on cause and duration: local, riverine, coastal, and flash floods. A flood warning system should be developed according to the target flood type. For instance, the European Flood Forecasting System (EFFS) was developed to forecast 10 days prior to riverine floods for the major rivers in Europe





(Pappenberger et al., 2005; Thielen et al., 2009). The U.S. Geological Survey Caribbean Weather Science Center developed the Real-Time Flood Alert System (RTFAS) for the early detection of flash floods (López-Trujillo, 2010). For low-lying areas threatened by storm surges, coastal flood warning systems have also been developed (De Kleermaeker et al., 2012; Doong et al., 2012). The present study developed an effective early warning system to

estimate urban inundation risk caused by high-intensity, short-duration rainfall.

Various approaches are used to simulate flooding, given the available rainfall data. Complex models such as the one- or two-dimensional Saint-Venant equations better describe flow behaviors and provide detailed spatial information as part of their flood forecasts (e.g., Nguyen et al., 2015; Huthoff et al., 2015). However, high computation costs and substantial data requirements are involved in solving these models' detailed governing equations. The efficiency and

numerical stability of warning systems are important issues that may limit the models' applications during an emergency response or real-time forecast. Thus, a variety of alternatives have been developed to improve the models' computing efficiency. Three approaches have attracted the most attention: simplified, equations-based systems; data-driven models; and rainfall threshold-based approaches. Because solving complex governing equations such as conservation of mass, momentum, and energy is extremely time consuming, many studies now utilize simplified

equations such as Manning's equation to describe water spreading (e.g., Cirbus and Podhoranyi, 2013; Liu et al., 2014; Shao et al. 2015). These studies divide the modeling domain into a lattice of cells and utilize Manning's equation to calculate water's spreading velocity. The ratio between cell size and velocity is the time-step iteration. Water exchange between cells can then be determined by taking the product of the time-step iteration and velocity. This approach has improved the calculation efficiency of forecasting models and provides acceptable results. However, the data required,

including digital elevation models (DEMs) and surface roughness, are sometimes difficult to collect. As a result, data preparedness is still a practical concern for the abovementioned models. Because a high-resolution DEM is necessary to provide accurate results in the abovementioned models, a large number of cells with a very small time-step iteration are created in the modeling domain. As a consequence, generating long-duration forecasts still results in low computational efficiency. Data-driven models are usually based on computational intelligence or machines. They

usually involve mathematical equations derived from the analysis of time series data and have multiple applications. Flood forecasting is just one of these applications (e.g., Chang et al., 2010; Lin et al., 2013). For example, Chang et al. (2010) developed a regional flood inundation forecast model to provide a flood inundation map with a lead time of 1 h. The model is composed of linear regression models and artificial neural networks. As indicated by the name, the quality and quantity of data used in the model have a considerable impact on the performance of data-driven models.

To collect accurate flood inundation data is a challenge in itself. In addition, the performance of data-driven models deteriorates as forecast time increases (e.g., Lin and Jhong, 2015; Badrzadeh et al., 2015). Data-driven models also cannot provide forecasts with longer lead times. A rainfall threshold approach is commonly applied to evaluate landslide risk (e.g., Crosta and Frattini, 2003; Guzzetti et al., 2007; Posner and Georgakakos, 2015). Meteorological organizations and civil protection agencies generally use rainfall thresholds to issue flood forecasts/warnings (Martina

et al., 2006). For example, the US National Weather Service (NWS) developed Flash Flood Guidance (FFG) values for flash flooding (Carpenter et al., 1999). Floods are predicted or flood warnings are issued if a critical value—namely, a rainfall threshold—is exceeded by the observed or forecasted rainfall. Georgakakos (2005, 2006) studied operational



flash flood warning systems based on FFG and provided analytical results. These studies found that an FFG threshold is likely to produce a high probability of detection in regions where flash floods are frequent. There are six FFG values—1, 3, 6, 12, and 24 h—and the NWS River Forecast Centers routinely issue flash flood warnings throughout the day for every county in the United States. Several operational meteorological agencies throughout the world issue

warnings based on FFG values (Gourley et al., 2014). The European Flood Awareness System (EFAS) uses numerical weather predictions and the European Precipitation Index, which is based on simulated climatology, an FFG-related concept, to provide flash flood warnings. For Kenya (Hoedjes et al, 2014), Haiti (Shamir et al.,2013) and other developing countries (Georgakakos et al., 2013) that do not have enough well-trained operators and sources to set up an efficient flood warning system, the approach is a viable alternative that allows for the mitigation of flood damage.

After all, the rainfall threshold approach's ability to produce rapid flood risk assessment at the national level has been clearly demonstrated. It has proven successful in identifying a number of flash floods across Europe (Alfieri et al., 2014). Although it should not be considered a substitute for complex hydro-meteorological models because of its simplicity, using a rainfall threshold approach to develop a flood warning system can be an immediately useful tool for a variety of decision makers interested in early warnings and flash floods (Martina et al., 2006).

This study integrates a rainfall threshold approach and quantitative precipitation forecasts (QPFs) to provide a practical urban inundation warning system. By directly comparing QPFs with critical rainfall thresholds, this study aims to propose an early warning system that provides forecasts, allows for the possibility of issuing urban inundation warnings and gives response agencies enough lead time to implement emergency preparedness plans. Compared to in situ observation networks such as rainfall gauges and radar, the QPFs generated by numerical weather models can

extend forecasting lead times. Consequently, a flood warning system that uses QPFs as the rainfall input could increase the forecasting horizon from a few hours to a few days (Pappenberger et al., 2005; Shi et al., 2015). Georgakakos (2005) concluded that the dominant source of uncertainty in applying a rainfall thresholds approach to evaluate flood risk is precipitation. As a model input, the uncertainty in forecasted rainfall values is generally higher than that for observed rainfall data. Nevertheless, to extend the forecast lead time, operational and research flood forecasting

systems around the world are increasingly moving toward using QPFs to provide early warnings (Cloke and Pappenberger, 2009). Martina et al. (2006) discussed the possibility of providing flood warnings at given river reaches by directly comparing the QPF to a critical rainfall threshold value. Shamir et al. (2013) integrated FFG and QPF data to provide 36-h forecasts of flash flood occurrences during the passage of Hurricane Thomas in Haiti. Most of the abovementioned studies applied rainfall thresholds in flash/riverine flood forecasting. Only a few studies (Jang, 2015;

Wu et al., 2015) have applied rainfall thresholds to evaluate urban inundation risk. The present study's use of the rainfall thresholds approach and QPFs to evaluate inundation risk is the first attempt of its kind in Taiwan. Regardless of the forecasts' uncertainty, considering which probabilistic forecast levels should be used to issue inundation alerts or take actions is a challenging topic. Higher levels of probabilistic forecasts usually give the practitioner more confidence in the results. Dale et al. (2014) proposed a risk-based decision-support framework that could be easily

applied in an operational flood forecast and early warning context. Other studies have also discussed the selection of appropriate probabilistic forecasts in terms of the economic and practical consequences of taking action (Coughlan de Perez et al., 2015; Coughlan de Perez et al., 2016). Therefore, the present study evaluates the system's performance

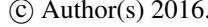



in terms of different levels of forecast probability. Finally, to decrease the uncertainty from rainfall forecasts and improve the model's inundation forecasts, this study proposes a data assimilation technique that uses real-time observations to modify rainfall forecasts and increases the 24-h forecast accuracy. The remainder of this paper is organized as follows. Section 2 describes the system's development, including the QPFs, rainfall thresholds and inundation risk evaluation process, as well as a data assimilation technique to increase the forecast's accuracy. Section 3 briefly describes the study area and the data used in the study. Sections 4 and 5 present the results and conclusions.

## 2 System development

The proposed early inundation warning system integrates ensemble precipitation forecasts, rainfall thresholds, and a real-time data assimilation technique to assess the possibility of issuing inundation alerts. Figure 1 shows the system's operational process during a typhoon event. The forecast results are intended to be provided to practitioners through a webpage. Due to a limitation in the computing resources and data retrieval tools available, the system generates a forecast every 6 h and updates the results on the webpage. The details of each component in the system are as follows.

### 2.1 Ensemble precipitation forecasts for system input

This study used rainfall forecasts from a precipitation ensemble forecast experiment, namely, the Taiwan cooperative precipitation ensemble forecast experiment (TAPEX). TAPEX is a collective effort among academic institutes and government agencies such as National Taiwan University (NTU), National Central University (NCU), National Taiwan Normal University (NTNU), Chinese Culture University (CCU), the Central Weather Bureau (CWB), the National Center for High-Performance Computing (NCHC), the Taiwan Typhoon and Flood Research Institute (TTFRI), and the National Science and Technology Center for Disaster Reduction (NCDR). The experiment began in 2010 and was the first attempt to design a high-resolution numerical ensemble weather model in Taiwan. The experiment collects worldwide observation data, including temperature, wind, surface pressure, and relative humidity, from satellites, atmospheric sounding devices, buoys, aviation routine weather reports, ships, and other available sources (e.g., Hsiao et al., 2012; Hsiao et al., 2013). TAPEX uses the outputs from the Global Forecast System (GFS) produced by the National Centers for Environment Prediction (NCEP), along with observation data, as the initial and boundary conditions for its forecasts. Various model physics schemes and data assimilation strategies are used to perturb the numerical weather models and create differentiated ensemble members. To date, twenty ensemble members and four different regional models (AWR-WRF, HWRF, MM5 and CreSS) have been established for precipitation forecasting. The experiment aims to provide 24-, 48-, and 72-h typhoon rainfall forecasts and generates four runs per day at a 5-km spatial resolution. TAPEX's rainfall forecasts can extend the inundation forecast lead time to 72 h, which exceeds the average rainfall-runoff concentration time and the lag between observed peak participation and flooding in Taiwan. This lead-time is thus considered sufficient for decision-making processes to be implemented prior to inundation.





## 2.2 Rainfall threshold for urban inundation alerts

Coughlan de Perez et al. (2016) defined the danger level of flooding as the 95[th] percentile of a flood model's forecasts at a lead-time of 0 h. The present study considered rainfall thresholds as danger levels related to the likelihood of urban inundation. In Taiwan, the Water Resources Agency (WRA) has developed rainfall thresholds for all townships

(Wu and Wang, 2009). Inundation alerts are issued when observed rainfall meets or exceeds a given rainfall threshold. Local governments and civil agencies take necessary measures such as evacuating residents and deploying dewatering pumps based on the alerts. Given the historical record, the WRA assumes that inundations are directly related to accumulated rainfall and use a regression analysis to identify a two-level alarm for five duration periods. The five duration periods are 1, 3, 6, 12, and 24 h; a total of 10 rainfall thresholds are used to issue urban inundation alerts. The

two levels of alarms are defined as follows:

**First-level alert**: If the rain continues, the roads and villages subject to a high risk of flooding in the alerted townships may flood.

**Second-level alert**: If the rain continues, the roads and villages subject to a high risk of flooding in the alerted townships will flood in the next 3 h.

The WRA has associated different rain gauges with different townships and issues warnings by comparing the observations with the associated rain gauges. An inundation alert is issued if any of the rainfall thresholds is met by the observed rainfall. Wu (2013) compared the alerts to collected inundation records in 2012 and 2013 and concluded that the forecast accuracy rate is above 60%. As the only rainfall thresholds approach used to issue inundation alerts in Taiwan, it has proven its applicability in predicting flood inundation.

## 2.3 Inundation risk evaluation and a data assimilation technique to modify the forecasts

In practice, the WRA issues inundation alerts when the cumulative rainfall exceeds the rainfall threshold at time T (Figure 2). However, WRA compares real-time precipitation observations to the rainfall thresholds, and thus the lead time is usually not long enough to allow communities to implement emergency preparedness measures. This study proposes a practical early warning system that compares cumulative projected rainfall instead of observed rainfall to

provide probabilistic urban inundation forecasts. The system uses TAPEX's forecasted rainfall to extend the model's lead-time to 72 h. Figure 3 shows the forecast length during a real-time operation. TAPEX uses available observations at *t-6* as its model's initial conditions, and its numerical weather model computation process took 6 h to produce rainfall forecasts from t to *t+72* h. A total of 352 Taiwanese townships were used in this study to evaluate the proposed system's performance. Equations (1) and (2) were used to calculate the probability of inundation in any given township;

the forecasts were displayed over three distinct time periods (1-24 h, 25-48 h, and 49-72 h). A rolling window approach was applied to estimate the probability of issuing an inundation alert: each hour of the forecasting period was considered an evaluation end point, and the cumulative rainfall was calculated for the different durations.

$$f_i = \begin{cases} 1 & \text{if } PF_{i,accu} \geq PT_{dur} \\ 0 & \text{if } PF_{i,accu} < PT_{dur} \end{cases} \quad dur = \{1, 3, 6, 12, 24 \text{ h}\} \tag{1}$$





where $PF_{i,accu}$ is the cumulative forecasted rainfall of the $i^{th}$ ensemble member in TAPEX. $PT_{dur}$ represents cumulative rainfall thresholds for the different durations ($dur$) (1, 3, 6, 12, and 24 h). An inundation occurred ($f_i$ =1) if the cumulative forecasted rainfall exceeded any of these thresholds.

$$Pr = \frac{1}{N} \sum_{i=1}^{N} f_i \times 100, N = 1, 2, \dots, 20 \qquad (2)$$

There are a total of 20 ensemble members ($N$=20) in TAPEX. Equation (2) sums the $f_i$ values to obtain a probability ($Pr$), which represents the inundation risk for any given township. Each township's inundation risk can be obtained by repeating the above steps and comparing the results to TAPEX's 72-h rainfall forecasts. Three separate time periods (1-24 h, 25-48 h, and 49-72 h) illustrate the township's future inundation risk.

   The accuracy of the rainfall forecasts has a considerable impact on the flood inundation forecasts. The complexity of

earth-atmosphere system and associated physical interactions adds uncertainty to the ensemble rainfall forecasts. To decrease the uncertainty in the rainfall forecasts, this study used real-time rainfall observations to modify the rainfall forecasts and improve the 24-h urban inundation forecast's performance. Figure 3 illustrates the combination of observations and forecasts used in the forecasting process. This study utilized five rainfall thresholds to represent different rainfall durations. However, these five thresholds could not be applied to evaluate the inundation risk at every

hour within the first 24-h forecast. For example, only one rainfall threshold covers the 1-h period, which can be considered time $t$ in Figure 3; however, there is a lack of forecasts for $t$ - 1 and the preceding hours. When $t = t + 2$, only rainfall thresholds for 1 and 3 h can be adopted. This shortcoming results in the underestimation of inundation forecasts. Given the above assumption, all five duration periods are applicable after the 25th h. This study proposes a data assimilation technique using observed rainfall data to address the absence of rainfall forecasts. It applies available

observation data from t - 24 to t - 1 prior to issuing inundation forecasts at $t$ (Figure 3). Figure 2 combines the observation data (red line) and forecasts (dash line) with all rainfall thresholds (solid blue line). Alerts are issued if the combination exceeds the rainfall threshold at any given duration. In other words, the inundation forecast is improved within the first 24 h.

## 3    Study area and data

### 3.1    Study area

   Taiwan has an area of approximately 36,000 km$^2$, and approximately 70% of the island is covered by mountains. A mountain range runs through the center of the island from north to south and forms a ridge dividing the east- and west-bound rivers. The rest of the island is composed of alluvial plains below 100 m in elevation. Ninety percent of the population lives on these alluvial plains. The distance from the mountaintops to the sea is very short, less than 70 km

on average. Most of the riverbed slopes exceed 1/100 in the upstream reaches and are between 1/200 to 1/500 in the downstream reaches, which results in average rainfall-runoff concentration times of between 6 and 72 h in the townships (Jang, 2015) and a lag time between observed peak precipitation and flooding of between 2 and 10 h (Jang et al., 2012). Taiwan is situated on one of the primary paths for western North Pacific typhoons and is affected by an





average of 3.4 typhoons each year. Taiwan's average annual rainfall is 2,600 mm, which is 2.5 times greater than the global average; and 80% of the precipitation on the island is caused by typhoons and storms from May to October (Cheng and Liao, 2011). Typhoons bring heavy rainfall and cause severe floods in Taiwan. The short concentration time and high density of the population in the plains areas further increase the damage caused by floods. Taiwan is

one of the most disaster-prone countries in the world; thus, it has been selected as the study area here for the development of an urban inundation warning system.

### 3.2    Observed inundation alerts

Records such as the time of occurrence, depth, and extent of inundation are used to calibrate and validate early warning systems. Collecting accurate information is thus incredibly important. However, data collection during major floods

is challenging. For example, identifying the occurrence time of an inundation is always an issue because of the lack of in situ monitoring devices. This study used urban inundation alerts issued by the WRA as a reference to evaluate the system's performance. The WRA issues alerts following the Common Alerting Protocol (CAP), which was first published by the OASIS Emergency Management Technical Committee in 2005 (OASIS, 2005). The WRA updates its alerts every ten minutes and uploads the information to an open-source platform operated by the National Science

and Technology Center for Disaster Reduction (Lee et al., 2014). The CAP data include observed flood warning information, such as the flood warning's location and duration. Information on seven typhoons, including SOULIK (2013), TRAMI (2013), MATMO (2014), FUNG-WONG (2014), LINFA (2015), SOUDELOR (2015), and DUJUAN (2015), was collected to evaluate the system's performance. Five of these typhoons made landfall and resulted in heavy rainfall and floods. For example, SOUDELOR dropped more than 1,100 mm of precipitation within 24 h and

had wind gusts of up to 66.1 ms$^{-1}$ in northern Taiwan (i.e., Suao Township, Yilan County). Detailed information on these seven typhoons is listed in Table 1. The landfall time was identified when the eye of the typhoon made landfall. Of these typhoons, the eyes of TRAMI and LINFA did not make landfall. For reference, this study selected the minimum observed atmospheric pressure at a weather station to define the time when these two typhoons were closest to Taiwan. The selected weather stations were the Taipei station for TRAMI and the Kaohsiung station for LINFA.

## 4    Results and discussion

This study relied on the contingency information shown in Table 2 to evaluate the performance of the proposed system. Hits and misses were associated with the observed records and determined based on whether the system's warning forecasts were consistent with the observations. A false alarm was associated with forecasts that did not correlate with observed data. "No event" was assigned to a township when neither the CAP records nor the model indicated flooding.

Because floods are not frequent events, the no event (no flooding) scenario typically had a higher frequency than the other three fields. Different measures that have been broadly adopted by previous studies (e.g., Nguyen et al., 2015; Yang et al., 2015; Zhang et al., 2015) were used to evaluate the system's performance:

$$Pobability\ of\ detection\ (POD) = \frac{Hit}{Hit+Miss},$$    (3)





$$False\ alarm\ ratio\ (FAR) = \frac{False\ alarm}{Hit+false\ alarm},$$ (4)

$$Success\ ratio\ (SR) = \frac{Hit}{Hit+false\ alarm},$$ (5)

$$Threat\ score\ (TS) = \frac{Hit}{Hit+Miss+false\ alarm}.$$ (6)

Both *POD* and *TS* are sensitive to hits and range from 0 to 1. The only difference between these two values is that

*POD* ignores false alarms and *TS* does not. *POD* has the ability to be artificially improved by the issuance of additional

alarms, which would increase the number of hits. *TS* is also known as the critical success index (*CSI*) and usually

results in poorer scores for rare events. *SR* and *FAR* are the success ratio and false alarm ratio, respectively. *FAR* is

used in conjunction with *POD*. If *FAR* equals 0.5 or less, the performance is considered tolerable (Coughlan de Perez

et al., 2016). The sum of *SR* and *FAR* equals 1, and both indices ignore misses. This study combined *SR* and *FAR* into

one index (*SR-FAR*) that had a range from -1 to 1. A positive value (> 0) for *SR-FAR* was expected given that the

likelihood of correct warnings is acceptable. Rare events such as floods result in extremely large numbers of no events,

which could greatly affect the forecast results. In this study, a no event forecast can provide information to decision

makers that allows them to allocate resources to those townships with a higher inundation risk. Equations (3) to (6)

do not consider the "no event" scenario in their formulas. The accuracy (*ACC*) of the model, which is shown in

Equation (7) and is also called the proportion of correct forecasts (Wilks, 2005), is simple and intuitive, and it served

as a valuable reference in this study.

$$Accuracy\ (ACC) = \frac{Hit+No\ event}{Hit+False\ alarm+Miss+No\ event}$$ (7)

The next section presents the performance evaluation of the proposed system and then modifies the forecasting results

using a hybrid of real-time observation and rainfall forecasts to improve the first 24-h inundation forecasts. This study

used the time the typhoon made landfall as a reference point to define the evaluation period. The time needed to

generate a rainfall forecast is 6 h, noted as one date-time group (dtg). The evaluation period was plus-minus three dtgs

(18 h) relative to the time at which a typhoon made landfall. The average impact duration of a typhoon in Taiwan is

73.68 h (Huang et al., 2012). A typhoon has the most impact during the evaluation period (a total of 36 h).

### 4.1 Comparisons of forecasted results without a data assimilation technique

Both typhoon tracks and geography impacted the performance of the rainfall forecasts. Figure 4 shows the observed

typhoon tracks, and Figure 5 compares the forecasted and observed tracks for SOULIK, SOUDELOR, and MATMO.

The models of the first two typhoons were consistent with the observed tracks, while the third was not; as a result, the

performance of rainfall forecasts during the first two typhoons exceeded that of the third typhoon. The causes of the

track forecast errors are beyond the discussion of this study. Use of ensemble rainfall forecasts as inputs to produce

flood warning forecasts should take into account uncertainties such as track and rainfall forecast errors in numerical

weather predictions. Figures 6-8 show the differences between the observed and forecasted flood warnings over three

lead-time periods (1-24 h, 25-48 h, and 49-72 h). Tables 3 to 5 summarize the average *ACC*, *POD* and *SR-FAR* results

for different lead-time lengths during the evaluation period. The proposed system provides probabilistic forecasts. For

example, 50% flood probability means that at least 10 out of 20 TAPEX members produced rainfall forecasts that met




or exceeded the rainfall thresholds. The appropriate probability threshold that initiated response actions was discussed. Six probability thresholds (10%, 30%, 50%, 70%, 80%, and 100%) were selected. The results showed that forecasts with lower possibility thresholds had higher *TS* scores (Figures 6-8). For example, Figure 6 shows that the *TS* scores of SOUDELOR are 0.1-0.4 for the 10% probability threshold, which are higher than those for the 70% probability

threshold. All tables showed that the average performance of low-possibility thresholds over the evaluation period resulted in better *TS* and *POD* scores. A lower probability threshold means a lower inundation threshold. Thus, the number of hits was increased and the number of false alarms was increased as well. Decision makers generally consider an increased number of actions "in vain" when taking emergency measures based on a low probability threshold. The higher probability thresholds (e.g., a probability threshold > 50%) had lower *TS* scores and  indicated

that TAPEX ensemble rainfall forecasts were usually under  estimated in this study. TAPEX's forecasted tracks had an impact on the rainfall forecasts, which affected the accuracy of the inundation forecasting. SOUDELOR and SOULIK had the best performance in terms of *TS* scores. The results for these typhoons were consistent with the track forecasts' performance (Figure 4). The results also showed that the *TS* performance decreased after the typhoons made landfall. The period from -3 dtg to landfall is shown in Figures 6-8. Taiwan's terrain has a significant impact on the

formation of a typhoon vortex in numerical weather models. The typhoons, due to their proximity to Taiwan by the time of model initiation, were not well developed in the models because of the terrain. Consequently, the typhoon tracks, rainfall, and related inundation forecasts were inevitably influenced. In the tables, the majority of *ACC* values exceeded 0.7. The less likely the inundation, the higher the *ACC* value. For example, only a few inundation alerts were issued during LINFA; the system's corresponding *ACC* scores were above 0.9. However, the *POD* and *SR-FAR* values

were not as good as the *ACC* values in this case. The *POD* scores were zero. The *SR-FAR* values could not be calculated because there were zero hits and false alarms. When the system produced less accurate forecasts, the performance of the *POD* and *SR-FAR* functions decreased, resulting in a lower number of observed inundation alerts. A large number of inundation alerts were issued by the WRA during SOUDELOR and SOULIK. The *ACC* numbers were below 0.8. The *POD* and *SR-FAR* numbers were relatively better than those in LINFA. A lower possibility

threshold indicated that more hits and false alarms occurred; this resulted in negative *SR-FAR* scores. In general, the *SR-FAR* scores decreased when the forecast lead time increased. However, the results for SOULIK were opposite for the 50% probability threshold and below. The *TS* score was higher when the probability increased by up to 50% prior to the typhoon making landfall (i.e., -1 dtg). The number of false alarms decreased when the probability threshold increased. This helped improve the *TS* score at -1 dtg. However, this finding did not hold true when the probability

threshold was above 70%. Typhoon MATMO performed worst in terms of *SR-FAR* scores for the three different lead-time lengths. Figure 5 shows that the forecasted tracks did not coincide with the observed track. When a typhoon making landfall, the topography had an impact on the performance of numerical weather models and it worsened the performance of the inundation warning forecasts. All of the results above indicate that the greatest uncertainty in the forecasts appears in the numerical weather predictions, which also has an important impact on other related disaster

forecasts.





### 4.2    Modified forecasts using the data assimilation technique

To decrease the uncertainty of numerical weather predictions and improve the performance of inundation alert forecasting, this study developed a hybrid real-time observed and forecasted rainfall model to improve the accuracy of early warning notifications. The longest rainfall threshold duration to trigger an inundation alerts is 24 h. The hybrid

technique was used to address the gap in forecasted rainfall data with observed rainfall information. The absence of forecasted rainfall values occurred in the first warning period (i.e., 1-24 h). Therefore, this study used the hybrid technique to improve the 1- to 24-h forecasts. Table 6 shows the modified forecast results compared to the original forecasts. Compared to the results without the hybrid technique, all performance measures' scores improved significantly. For example, when all typhoons were tested using the original forecasts, the system performed best

during SOULIK. Using the hybrid technique, the *POD* scores improved from 0.517 to 0.783 and from 0.002 to 0.245 for the 10% and 100% probability thresholds, respectively. The *TS* scores improved from 0.293 to 0.513 and from 0.002 to 0.235 for the 10% and 100% probability thresholds, respectively. The probability threshold represents the number of ensemble members' forecasted rainfall events that met or exceeded the rainfall thresholds. The hybrid technique forecasts thus support the idea that a higher probability threshold indicates lower uncertainty in terms of

forecasting. The *FAR* and *POD* scores decreased when the probability threshold increased. Decision-making confidence increases when the probability threshold increases and the *FAR* decreases. Coughlan de Perez et al. (2016) concluded that the likelihood of taking a necessary action when the *FAR* is lower than 0.5 would satisfy the decision maker's requirements for not taking action potentially in vain. Table 6 shows that most of the *FAR* scores improved to below 0.5 using the hybrid technique. Though these values improved compared to previous results, all of the *POD*

scores were still low and continued to decrease when the probability threshold increased. The low *POD* score implies a lower hit rate. To improve these values, identifying the accuracy and uncertainty of rainfall forecasts is necessary.

### 5    Conclusions

This study proposed an early inundation warning system that integrates ensemble rainfall forecasts and rainfall thresholds. Five rainfall thresholds with different durations were applied. Seven typhoon events during the period

2013-2015 and real inundation alert records from the WRA were used to evaluate the model's performance. Five performance measures and a period of 18 h before and after a typhoon made landfall were considered. The system applied ensemble rainfall forecasts and provided probabilistic forecasts. Therefore, six different probability thresholds were considered to trigger the issuance of inundation alerts and calculate various performance scores. An appropriate probability threshold helps decision makers take fewer actions in vain. The results showed that a lower probability

threshold had a higher *POD* score, which is associated with a higher inundation alert detection rate. The downside of a lower probability threshold is a higher *FAR* score. If the *FAR* is above 0.5, the system is considered impractical (Coughlan de Perez et al., 2016). In conclusion, this study was unable to identify the most useful probability threshold for identifying when emergency responders should take various actions. Numerical weather predictions were the dominant input influencing the forecast results. The model's performance varied according to the different typhoons

tested. For example, the system performed best in terms of forecasted tracks and rainfall during SOUDELOR and





SOULIK of all the typhoons. The system also performed best in terms of forecasting inundation from these two typhoons.

This study evaluated the system's forecast results based on typhoon locations. Using the time a typhoon made landfall as a reference point, the system performed better before a typhoon made landfall, particularly in terms of *TS* scores.

Taiwan's steep terrain poses a challenge to the vortex initialization in numerical weather prediction models. Most current techniques are unable to properly initiate a typhoon vortex near complex terrain, when in reality the typhoons were already well-developed at the time of landfall, which impacted their tracks, rainfall, and associated inundation forecasts. As a result, terrain contributes to the uncertainty inherent in using numerical weather prediction models, particularly in Taiwan. Based on these results, the authors were unable to identify an appropriate probability threshold

to enable decision makers to take fewer emergency response actions in vain. This study's findings suggest that a better forecast is usually produced (1) when the forecasted typhoon tracks are consistent with the observed tracks and (2) before a typhoon makes landfall.

Finally, the authors developed a data assimilation technique that combined real-time observed and forecasted rainfall to decrease the uncertainty of numerical weather predictions and to improve inundation forecasts. The concept used

observed rainfall to fill in gaps in forecasted rainfall data so that all five rainfall thresholds could be considered within the first 24-h period. The results showed that all five performance measures were significantly improved by using this hybrid approach. The *FAR* scores decreased when the probability threshold increased. All *FAR* scores were below 0.5 or less when the probability threshold was 30% or above. This technique improved the appeal of the early warning system and generated more valuable forecasts that allowed decision makers to take fewer actions in vain. To further

decrease the uncertainty of numerical weather predictions and improve the performance of inundation forecasts, advanced techniques such as radar observations and associated data assimilation systems could be applied. A greater number of extreme weather events will appear in the future due to global climate change. These extreme events will bring high-intensity rainfalls over very short time spans. Radar observations efficiently improve very short-range rainfall forecasts, which are essential for accurate inundation forecasts. Rainfall thresholds need to be updated to meet

the present flood capacity, such as when a new storm sewage system is put in place. After all, decision makers use forecasted rainfall and threshold-based early warning systems for a high-level overview of flood risk only. Given its advantage of an extended lead time and rapid estimation process, the model presented here is beneficial for emergency deployment to prepare large areas in advance of flooding. For small area forecasts during a disaster, a complex physics-based model is recommended to replace the threshold-based model and provide detailed information.






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

**Acknowledgement**





The authors would like to thank the Water Resources Agency, Taiwan for providing the rainfall threshold values and the Central Weather Bureau, Taiwan for providing rainfall observation data.




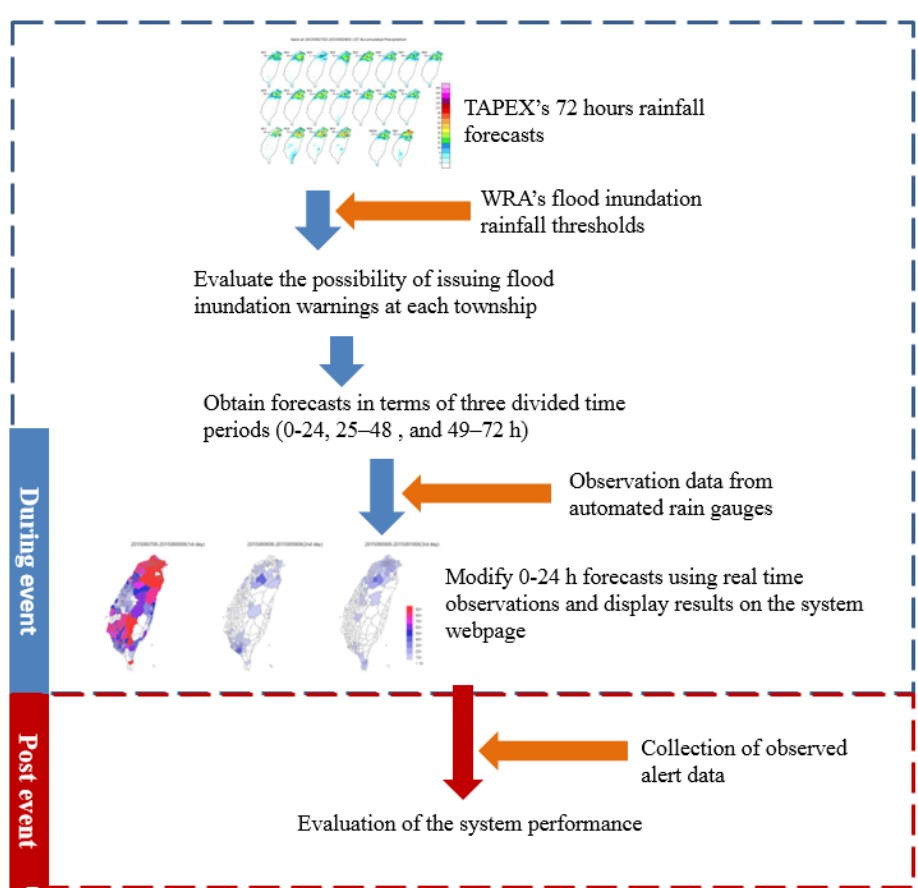

**Figure 1: The operational flow chart for the proposed urban inundation early warning system.**

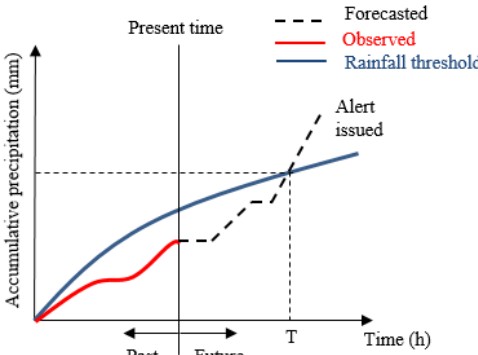

**Figure 2: WRA issues an inundation alert when observed rainfalls meet or exceed any given rainfall**
5   **thresholds (Modified from Martina et al., 2006).**





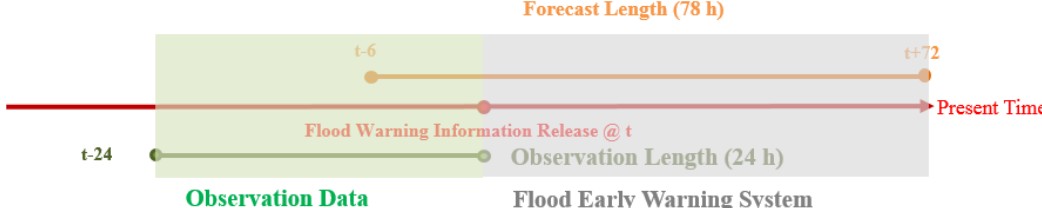

**Figure 3: A combination of real time rainfall observations and forecasts to improve 1- to 24-h inundation forecasts.**

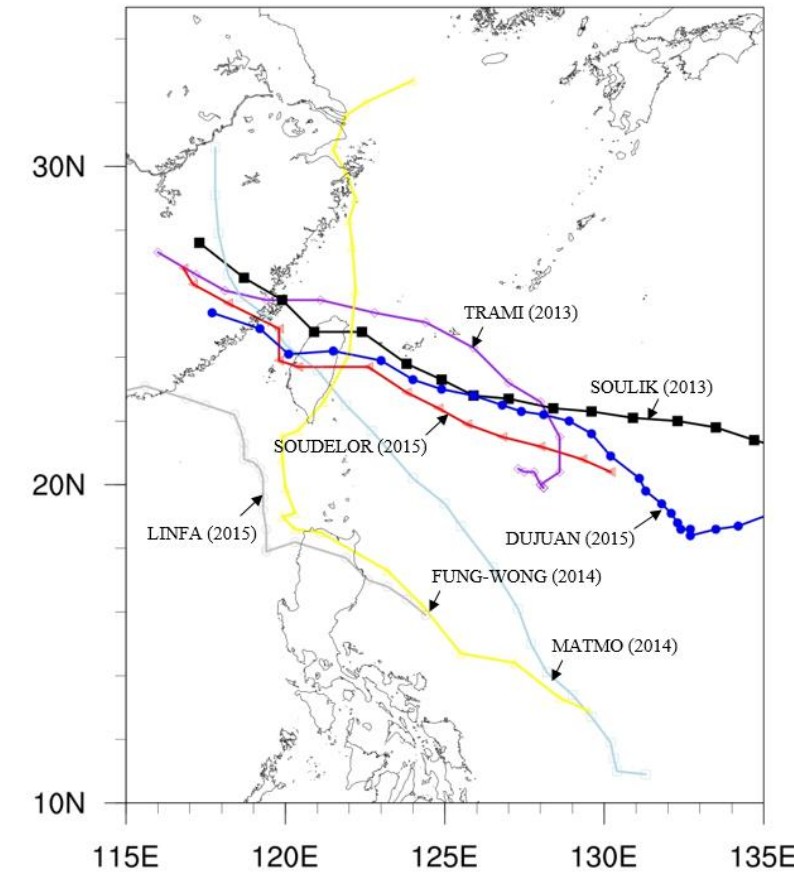

5          **Figure 4: Location of Taiwan Island, seven typhoons during 2013-2015, and their observed tracks.**





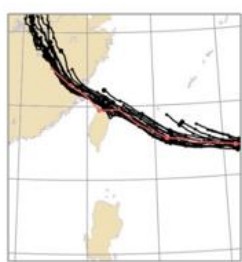
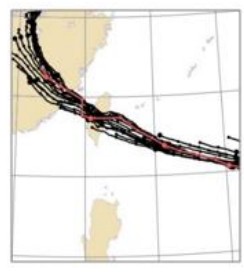
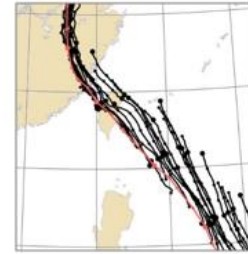

**Figure 5: Comparisons of Forecasted and observed typhoon tracks for SOULIK (left), SOUDELOR (middle), and MATMO (right): black lines are TAPEX's ensemble mean forecasted tracks and each black line's forecasting length is 72 h; red lines are observed tracks.**



**Figure 6: Comparisons of TS performance with a 1- to 24-h lead time considering various probability thresholds.**





25

**Figure 7: Comparisons of TS performance with a 25- to 48-h lead time considering various probability thresholds.**

30



**Figure 8: Comparisons of TS performance with a 49- to 72-h lead time considering various probability thresholds.**





Table 1: Information of seven typhoons during 2013 - 2015 used to evaluate the system performance.

| Year | Name | Warning period (LST) | Landfall time (LST) |
|------|------|---------------------|---------------------|
| 2013 | SOULIK | 2013/07/11 0830-2013/07/13 2330 | 2013/07/13 0300 |
| 2013 | TRAMI | 2013/08/20 1130-2013/08/22 0830 | 2013/08/21/ 1800* |
| 2014 | MATMO | 2014/07/21 1730-2014/07/23 2330 | 2014/07/23 0010 |
| 2014 | FUNG-WONG | 2014/09/19 0830-2014/09/22 0830 | 2014/09/21 1000 |
| 2015 | LINFA | 2015/07/06 0830-2015/07/09 0530 | 2015/07/08 1500* |
| 2015 | SOUDELOR | 2015/08/06 1130-2015/08/09 0830 | 2015/08/08 0440 |
| 2015 | DUJUAN | 2015/09/27 0830-2015/09/29 1730 | 2015/09/28 1740 |

* For typhoons did not make landfall, this study defined the landfall time while the minimum observational station pressure was observed when typhoon was closest to Taiwan.

Table 2: Contingency table used for the system performance evaluation.

| | | CAP records from WRA | |
|---|---|---|---|
| | | Issued | Not Issued |
| Forecasted by the | Issued | Hit | False alarm |
| proposed system | Not issued | Miss | No event |





**Table 3: Average performance with 1- to 24-h lead time during the evaluation period for all possibility thresholds.**

| % | SOULIK | | | TRAMI | | | MATMO | | | FUNG-WONG | | | LINFA | | | SOUDELOR | | | DUJUAN | | |
|---|---|---|---|---|---|---|---|---|---|---|---|---|---|---|---|---|---|---|---|---|---|
| | ACC | POD | SR-FAR | ACC | POD | SR-FAR | ACC | POD | SR-FAR | ACC | POD | SR-FAR | ACC | POD | SR-FAR | ACC | POD | SR-FAR | ACC | POD | SR-FAR |
| 10 | 0.787 | 0.517 | 0.033 | 0.776 | 0.186 | -0.433 | 0.778 | 0.241 | -0.626 | 0.914 | 0.348 | -0.341 | 0.976 | 0.196 | -0.419 | 0.758 | 0.396 | -0.050 | 0.839 | 0.252 | 0.083 |
| 30 | 0.795 | 0.201 | 0.204 | 0.823 | 0.021 | -0.250 | 0.867 | 0.046 | -0.472 | 0.944 | 0.219 | 0.360 | 0.981 | 0.000 | n/a* | 0.792 | 0.166 | 0.462 | 0.844 | 0.100 | 0.519 |
| 50 | 0.787 | 0.092 | 0.220 | 0.827 | 0.009 | 0.333 | 0.874 | 0.007 | -0.667 | 0.944 | 0.129 | 0.818 | 0.981 | 0.000 | n/a* | 0.780 | 0.077 | 0.517 | 0.834 | 0.027 | 0.222 |
| 70 | 0.781 | 0.031 | 0.133 | 0.826 | 0.000 | n/a* | 0.876 | 0.000 | -1.000 | 0.941 | 0.058 | 1.000 | 0.981 | 0.000 | n/a* | 0.775 | 0.049 | 0.514 | 0.832 | 0.007 | -0.143 |
| 80 | 0.779 | 0.015 | -0.059 | 0.826 | 0.000 | n/a* | 0.877 | 0.000 | -1.000 | 0.940 | 0.039 | 1.000 | 0.981 | 0.000 | n/a* | 0.772 | 0.031 | 0.500 | 0.832 | 0.000 | -1.000 |
| 100 | 0.780 | 0.002 | 1.000 | 0.826 | 0.000 | n/a* | 0.877 | 0.000 | n/a* | 0.937 | 0.000 | n/a* | 0.981 | 0.000 | n/a* | 0.767 | 0.005 | 0.000 | 0.833 | 0.000 | n/a* |

* n/a means that either *FAR* or *POD* had zero values in the denominator and cannot be calculated.

**Table 4: Average performance with 25- to 48-h lead time during the evaluation period for all probability thresholds.**

| % | SOULIK | | | TRAMI | | | MATMO | | | FUNG-WONG | | | LINFA | | | SOUDELOR | | | DUJUAN | | |
|---|---|---|---|---|---|---|---|---|---|---|---|---|---|---|---|---|---|---|---|---|---|
| | ACC | POD | SR-FAR | ACC | POD | SR-FAR | ACC | POD | SR-FAR | ACC | POD | SR-FAR | ACC | POD | SR-FAR | ACC | POD | SR-FAR | ACC | POD | SR-FAR |
| 10 | 0.732 | 0.746 | -0.127 | 0.621 | 0.494 | -0.544 | 0.620 | 0.297 | -0.779 | 0.913 | 0.374 | -0.337 | 0.923 | 0.271 | -0.856 | 0.756 | 0.616 | -0.038 | 0.842 | 0.383 | 0.075 |
| 30 | 0.821 | 0.453 | 0.258 | 0.817 | 0.117 | -0.187 | 0.811 | 0.086 | -0.757 | 0.947 | 0.245 | 0.490 | 0.978 | 0.063 | -0.625 | 0.796 | 0.291 | 0.265 | 0.833 | 0.053 | 0.023 |
| 50 | 0.825 | 0.274 | 0.594 | 0.830 | 0.044 | 0.357 | 0.857 | 0.023 | -0.781 | 0.944 | 0.116 | 0.895 | 0.981 | 0.000 | -1.000 | 0.787 | 0.147 | 0.400 | 0.832 | 0.010 | -0.111 |
| 70 | 0.795 | 0.081 | 0.725 | 0.827 | 0.009 | 0.333 | 0.877 | 0.010 | -0.143 | 0.940 | 0.039 | 1.000 | 0.982 | 0.000 | n/a* | 0.772 | 0.047 | 0.256 | 0.831 | 0.000 | -1.000 |
| 80 | 0.784 | 0.029 | 0.600 | 0.826 | 0.007 | 0.200 | 0.877 | 0.007 | 0.333 | 0.937 | 0.000 | n/a* | 0.982 | 0.000 | n/a* | 0.769 | 0.028 | 0.143 | 0.831 | 0.000 | -1.000 |
| 100 | 0.779 | 0.000 | -1.000 | 0.826 | 0.000 | n/a* | 0.877 | 0.000 | n/a* | 0.937 | 0.000 | n/a* | 0.982 | 0.000 | n/a* | 0.769 | 0.007 | 1.000 | 0.833 | 0.000 | n/a* |

* n/a means that either *FAR* or *POD* had zero values in the denominator and cannot be calculated.

5        **Table 5: Average performance with 49- to 72-h lead time during the evaluation period for all probability thresholds.**

| % | SOULIK | | | TRAMI | | | MATMO | | | FUNG-WONG | | | LINFA | | | SOUDELOR | | | DUJUAN | | |
|---|---|---|---|---|---|---|---|---|---|---|---|---|---|---|---|---|---|---|---|---|---|
| | ACC | POD | SR-FAR | ACC | POD | SR-FAR | ACC | POD | SR-FAR | ACC | POD | SR-FAR | ACC | POD | SR-FAR | ACC | POD | SR-FAR | ACC | POD | SR-FAR |
| 10 | 0.671 | 0.576 | -0.299 | 0.514 | 0.573 | -0.610 | 0.483 | 0.317 | -0.835 | 0.860 | 0.303 | -0.668 | 0.910 | 0.188 | -0.913 | 0.761 | 0.531 | -0.024 | 0.826 | 0.129 | -0.138 |
| 30 | 0.762 | 0.092 | -0.301 | 0.715 | 0.219 | -0.814 | 0.771 | 0.099 | -0.814 | 0.935 | 0.071 | -0.154 | 0.978 | 0.000 | -1.000 | 0.782 | 0.154 | 0.248 | 0.831 | 0.002 | -0.818 |
| 50 | 0.782 | 0.022 | 0.333 | 0.784 | 0.105 | -0.531 | 0.866 | 0.017 | -0.737 | 0.937 | 0.000 | -1.000 | 0.982 | 0.000 | n/a* | 0.768 | 0.040 | 0.022 | 0.831 | 0.000 | -1.000 |
| 70 | 0.780 | 0.000 | n/a* | 0.821 | 0.047 | -0.245 | 0.876 | 0.003 | -0.500 | 0.937 | 0.000 | n/a* | 0.982 | 0.000 | n/a* | 0.768 | 0.016 | 0.059 | 0.833 | 0.000 | n/a* |
| 80 | 0.780 | 0.000 | n/a* | 0.826 | 0.021 | 0.000 | 0.877 | 0.003 | 1.000 | 0.937 | 0.000 | n/a* | 0.982 | 0.000 | n/a* | 0.769 | 0.014 | 0.333 | 0.833 | 0.000 | n/a* |
| 100 | 0.780 | 0.000 | n/a* | 0.826 | 0.000 | n/a* | 0.877 | 0.000 | n/a* | 0.937 | 0.000 | n/a* | 0.982 | 0.000 | n/a* | 0.767 | 0.000 | n/a* | 0.833 | 0.000 | n/a* |

* n/a means that either *FAR* or *POD* had zero values in the denominator and cannot be calculated.

**Table 6: Average performance with 1- to 24-h lead time during the evaluation period for all probability thresholds using the hybrid technique**

| % | SOULIK | | | TRAMI | | | MATMO | | | FUNG-WONG | | | LINFA | | | SOUDELOR | | | DUJUAN | | |
|---|---|---|---|---|---|---|---|---|---|---|---|---|---|---|---|---|---|---|---|---|---|
| | POD | FAR | TS | POD | FAR | TS | POD | FAR | TS | POD | FAR | TS | POD | FAR | TS | POD | FAR | TS | POD | FAR | TS |
| 10 | 0.783 | 0.401 | 0.513 | 0.247 | 0.610 | 0.178 | 0.399 | 0.727 | 0.194 | 0.406 | 0.640 | 0.236 | 0.239 | 0.667 | 0.162 | 0.604 | 0.437 | 0.411 | 0.381 | 0.375 | 0.310 |
| 30 | 0.508 | 0.260 | 0.431 | 0.079 | 0.404 | 0.075 | 0.198 | 0.429 | 0.172 | 0.290 | 0.274 | 0.262 | 0.022 | 0.000 | 0.022 | 0.316 | 0.181 | 0.295 | 0.211 | 0.155 | 0.203 |
| 50 | 0.359 | 0.223 | 0.326 | 0.063 | 0.250 | 0.062 | 0.129 | 0.250 | 0.123 | 0.187 | 0.065 | 0.185 | 0.022 | 0.000 | 0.022 | 0.220 | 0.106 | 0.214 | 0.131 | 0.156 | 0.128 |
| 70 | 0.285 | 0.193 | 0.267 | 0.047 | 0.200 | 0.046 | 0.096 | 0.147 | 0.094 | 0.129 | 0.000 | 0.129 | 0.022 | 0.000 | 0.022 | 0.194 | 0.090 | 0.190 | 0.095 | 0.152 | 0.093 |
| 80 | 0.265 | 0.186 | 0.250 | 0.042 | 0.182 | 0.042 | 0.086 | 0.103 | 0.085 | 0.110 | 0.000 | 0.110 | 0.022 | 0.000 | 0.022 | 0.164 | 0.069 | 0.162 | 0.075 | 0.184 | 0.074 |
| 100 | 0.245 | 0.153 | 0.235 | 0.040 | 0.150 | 0.039 | 0.069 | 0.087 | 0.069 | 0.058 | 0.000 | 0.058 | 0.022 | 0.000 | 0.022 | 0.115 | 0.057 | 0.114 | 0.068 | 0.097 | 0.067 |

