# Peer review of "Using rainfall thresholds and ensemble precipitation forecasts to issue and improve urban inundation alerts"

_Hydrology and Earth System Sciences, 2016_

## Referee Comment (RC1) · Anonymous Referee #1 · 10 Aug 2016

The proposed manuscript presents the first application and evaluation of the TAPEX NWP ensemble to extend threshold based rainfall warnings of urban floods up to 72 hours over the entirety of Taiwan. It builds on previous work which had relied upon warnings from rain gauge observations which had consequently limited the maximum warning lead time.

The proposed method uses rainfall forecasts to identify the probability of exceeding 1, 3, 6, 12 and 24 hour rainfall thresholds for lead times of 1-24, 25-48 and 49-72 hours across 352 urban areas in Taiwan. Warnings are generated based on a probability exceedance level, experiments are made with different levels and evaluated against observed warnings disseminated during seven different typhoons. The results found

that higher threat score and probability of detection values were achieved when using lower probability thresholds, however this was at the cost of increasing the number of false alerts. In many cases the false alert ratio exceeded the success ratio meaning that the authors were unable to identify an exceedance probability level which could be used by emergency responders. The accuracy of the system was heavily driven by the accuracy of the forecasted typhoon track and showed better accuracy prior to the typhoon making landfall. The authors introduced observed rainfall data from the previous 24 hours in order that all five of the rainfall accumulation durations could be calculated during the first 24 hours of the forecast. This improved the results of all the skill scores during this period.

Overall this paper makes a good scientific contribution as it demonstrates the development of an urban flood warning system using NWP ensemble rainfall forecasts to extend the lead time. Forecasting urban flooding from intense rainfall events is of high importance but currently very few systems exist around the world. Therefore this manuscript will be of high interest to many readers who could apply the findings to help develop their equivalent warning systems. Therefore I recommend that this paper is accepted for publication after the authors have addressed the following minor corrections.

1. Provide more information about the rainfall thresholds It would be helpful for the authors to provide more information about how these rainfall thresholds are derived. A reference to Wu 2013 is provided but access to this conference proceeding does not seem possible, therefore the authors should provide more information within their manuscript. On page 5 at line 11 I do not understand the difference between the first and second level alerts, when is a second level alert created as opposed to a first level alert, does this have any impact on the evaluation presented in this manuscript?

2. Page 5 line 28 – do the 352 townships cover the entirety of Taiwan? This is a minor point but if they are only located in a certain part of the island then perhaps this could influence the evaluation scores?

3. Is there a mismatch between observed rainfall data and the TAPEX forecasts? If there is a bias in the TAPEX rainfall estimates, for example it could underestimate extreme precipitation, could this be problematic when comparing against rainfall thresholds derived from observations and when assimilating observations during day 1? If TAPEX does underestimate rainfall during a typhoon then this could mean that the threshold is not exceeded but the forecasted rain is still extreme when compared to the model climatology. On line 10 of page 9 the authors say that TAPEX rainfall forecasts were usually underestimated. In section 2.3 the authors should provide references to the verification scores of TAPEX forecasts of extreme rainfall and discuss the implications of comparing model forecasts against observed threshold values.

Does a similar issue also affect the assimilation of observation data during the first 24 hours? For example what would the affect on the results be if the authors assimilated data from the previous forecast as opposed to observation data? The authors should discuss this point either within section 2.3 or in their conclusions.

4. Are the data in figure 6 created using results with or without the data assimilation, this should be clarified in the text?

5. The explanation of the plotting of dtg in figures 6-8 needs clarification I am unclear about what is plotted at each dtg step in these figures, for example in figure 7 for cyclone Trami which made landfall on 20130821 1800 does this mean that the data plotted at 'landfall' refers to the 25-48 h data from the forecast on 20130819 1800, and the data plotted at -1 dtg refers to the 25-48 h data from the forecast on 20130819 1200 etc?

6. The proximity of a cyclone to landfall affecting the result accuracy requires further explanation On line 14 of page 9 it is suggested that when the TAPEX model is initialised with a typhoon close to or having already made landfall that this affects the track accuracy and hence the rainfall accuracy. I understand that this could be problematic for forecasts between 1-24 hours lead time but would this issue not diminish

with increasing lead time? For example the forecasts used in figure 8 (lead time 49-72 h), would these not have been initialised when the typhoons were yet to make landfall? My confusion here is linked to my confusion about the dtg in the previous point, so it would be greatly appreciated if the authors could provide clarification.

7. Provide a table of the skill scores calculated across all the typhoons This would give some useful overview statistics about the proposed system, it might also help the reader to determine how close the system is to providing a FAR<0.5 as alluded to by Coughlan de Perez et al., 2016 and at which threshold level.

The following are some minor typographical corrections:

Page 1 line 10 replace 'preceding' with 'predicting'

Page 1 line 31 replace 'European Flood Forecasting System (EFFS)' with 'European Flood Awareness System (EFAS)'

Page 3 line 6 the flash flood system in EFAS is currently the Enhanced Runoff Index based on Climatology (ERIC) see http://onlinelibrary.wiley.com/doi/10.1002/met.1469/abstract

Page 3 line 8 replace 'sources' with 'resources'

Page 4 line 8 replace 'early inundation warning system' with 'inundation early warning system'

Page 6 line 9 insert 'the' after 'The complexity of'

Page 7 equation 3 replace 'Pobability' with 'Probability'

Page 11 line 22 replace 'will appear' with 'are likely'

Figure 3 replace 'Present Time' with 'Timeline'

Table 1 caption insert 'that' so that the sentence reads 'For typhoons that did not...'

Table 5, in Matmo SR-FAR column for the 80% exceedance level did SR-FAR really

equal 1.0?

---

## Referee Comment (RC2) · Anonymous Referee #2 · 16 Sep 2016

In fact, the authors raised an essential point of research in their manuscript which is considered as (from my point of view) the highest priority for the hydrologists. I go through the paper several times in order to put my hand on how the authors introduced their contribution, I found that, the contribution is presented but not clear enough for me. Actually, while reading, there are several questions have been raised and also the difficulties in following the abbreviations reported in the text, so, first I recommend the authors to prepare a list of abbreviations at the beginning of the manuscript to easy following the manuscript. In addition, to avoid any duplication in the comments, I go through the first reviewer's report, I totally agreed with his comments and I am sure if the authors consider them, the manuscript will be in excellent shape for the readers.

[Figure]

However, I am highlighting hereafter some other comments as well. 1- The formulation of the rainfall threshold is not clear, showing comprehensive details on how they form it is essential for the readers 2- The introduction section is really very long, I suggest to shorten it to be direct to the point, to let the readers captured your idea in short way. Unless, the authors could split this section into several representative sub-sections. 3- The title of section 4-1 is not understandable 4- In section 4-2,,, at the beginning the authors start with the following statement "To decrease the uncertainty of numerical weather predictions and improve the performance of inundation alert forecasting, this study developed a hybrid real-time observed and forecasted rainfall model to improve the accuracy of early warning notifications.". This section supposed to be results and discussion section, but I did not see before that how the authors develop this hybrid real-time observation model. 5- Presentation of figures 5 and 6 are not of good quality. Figure 5 could be improved and increasing its scale. Figure 6, its notation is wrong (a, b) is repeated... and c the number 15 is appear which is not understandable for what. 6- Adding a paragraph at the end of discussion section showing the limitations of the proposed method would be very helpful for readers 7- The conclusion section is also very long and include several parts that consider as a discussion issue and not consider as conclusion, better to re-write this section to be direct to the point and reflect the objective of the study

---

## Author Comment (AC1) · 26 Oct 2016

Dear Reviewer #1:

We appreciate your precious comment. Please find the attached file for your reference.

Best wishes,

Josh

Please also note the supplement to this comment:
http://www.hydrol-earth-syst-sci-discuss.net/hess-2016-340/hess-2016-340-AC1-supplement.zip

---

## Author Comment (AC3) · 27 Oct 2016

Dear Editor:

The authors have replied the reviewers' comments and uploaded the modified manuscript. Thank you very much for your consideration.

Josh

Please also note the supplement to this comment:
http://www.hydrol-earth-syst-sci-discuss.net/hess-2016-340/hess-2016-340-AC3-supplement.zip

---

## Author Response (AR1)

**Ms. Ref. No.**: **doi:10.5194/hess-2016-340**

**Title**: Using rainfall thresholds and ensemble precipitation forecasts to issue and improve urban inundation alerts

**Responses to reviewer #1**

Overall this paper makes a good scientific contribution as it demonstrates the development of an urban flood warning system using NWP ensemble rainfall forecasts to extend the lead time. Forecasting urban flooding from intense rainfall events is of high importance but currently very few systems exist around the world. Therefore this manuscript will be of high interest to many readers who could apply the findings to help develop their equivalent warning systems. Therefore I recommend that this paper is accepted for publication after the authors have addressed the following minor corrections.

1. Provide more information about the rainfall thresholds. It would be helpful for the authors to provide more information about how these rainfall thresholds are derived. A reference to Wu 2013 is provided but access to this conference proceeding does not seem possible, therefore the authors should provide more information within their manuscript. On page 5 at line 11 I do not understand the difference between the first and second level alerts, when is a second level alert created as opposed to a first level alert, does this have any impact on the evaluation presented in this manuscript?

   Response:
   We appreciate for the Reviewer's comments. The rainfall thresholds for the first and second alerts are different. By definition, there is a 3-h lead time before flooding if the second-level alert is issued. The first-level alert is at an immediate risk of flooding. The WRA identified the rainfall thresholds of the second-level alerts for the purpose of precaution so the responding authorities have time to take associated actions. This study used the rainfall thresholds of the second-level alerts to evaluate the risk of flood alerts. Figure 2 is added to provide further information on the WRA's identification process of the rainfall thresholds. The text is also added in Section 2.2 in red. Please find the description on page 4 from lines 16 to 21 and lines 32 to 34 in the revised manuscript.

2. Page 5 line 28 – do the 352 townships cover the entirety of Taiwan?

This is a minor point but if they are only located in a certain part of the island then perhaps this could influence the evaluation scores?

Response:
We appreciate for the Reviewer's comments. Yes, the 352 townships cover the entirety of Taiwan Island. The evaluation score was for the whole island.

3. Is there a mismatch between observed rainfall data and the TAPEX forecasts? If there is a bias in the TAPEX rainfall estimates, for example it could underestimate extreme precipitation, could this be problematic when comparing against rainfall thresholds derived from observations and when assimilating observations during day 1? If TAPEX does underestimate rainfall during a typhoon then this could mean that the threshold is not exceeded but the forecasted rain is still extreme when compared to the model climatology. On line 10 of page 9 the authors say that TAPEX rainfall forecasts were usually underestimated. In section 2.3 the authors should provide references to the verification scores of TAPEX forecasts of extreme rainfall and discuss the implications of comparing model forecasts against observed threshold values. Does a similar issue also affect the assimilation of observation data during the first 24 hours? For example what would the affect on the results be if the authors assimilated data from the previous forecast as opposed to observation data? The authors should discuss this point either within section 2.3 or in their conclusions.

Response:

We appreciate for the Reviewer's comments. Yes, the underestimation of the rainfall forecast sincerely affects the flood forecasting. Mcbride and Ebert (2000) revealed that most numerical global models tend to over forecast the rainfall frequency on each threshold in Australia summer during December to February and slightly under forecast in winter. The study used a bias score to address the issue. If the bias score is smaller than 1.0, it means the rainfall forecast is underestimated. Mcbride and Elbert (2000) found that the bias score is below 1.0 when the rainfall threshold is 20 mm/day and above. According to the performance in 2016, it indicate that TAPEX under predicted the rainfall frequencies during a rainfall

event greater than 100mm/day. The bias scores were 0.49 to 0.12 for rainfall thresholds from 100 mm/day to 350 mm/day. The complexity of atmosphere gives the numerical weather forecasts a lot of uncertainties. However, that is beyond the scope of this study. The purpose of this study is to provide flood warning forecasts and improve the forecasts by adopting the uncertainties of numerical weather forecasts. The associated text is added in Section 2.3 accordingly in red. Please find the revised text from line 29, page 5 to line 8, page 6 in the revised manuscript.

4. Are the data in figure 6 created using results with or without the data assimilation, this should be clarified in the text?

Response:

We appreciate for the Reviewer's comments. The results in figure 6 are shown without the data assimilation technique. The section 4.1 is titled "Comparisons of forecasted results without a data assimilation technique" and figures 6-8 are in this section. To be clarified, the text "without a data assimilation technique" is added in the description of figures 6-8 and in the section 4.1. The title of the section 4-1 is changed to "Original forecast results without a data assimilation technique".

5. The explanation of the plotting of dtg in figures 6-8 needs clarification I am unclear about what is plotted at each dtg step in these figures, for example in figure 7 for cyclone Trami which made landfall on 20130821 1800 does this mean that the data plotted at 'landfall' refers to the 25-48 h data from the forecast on 20130819 1800, and the data plotted at -1 dtg refers to the 25-48 h data from the forecast on 20130819 1200 etc?

Response:

We appreciate for the Reviewer's comments. The Reviewer is correct. The landfall time is identified in Table 1. Taking TRAMI as an example, the landfall time is 20130821 1800 and the landfall dtg is 2013082114 for 1-24 h, 2013082014 for 25-48 h, and 2013081914 for 49-72h. Therefore, the -1 dtg is 2013082108 for 1-24 h, 2013082008 for 25-48 h, and 2013081908 for 49-72h. The text is added for further explanation. Please find the text on page 8 from lines 16 to 19.

6. The proximity of a cyclone to landfall affecting the result accuracy requires further explanation On line 14 of page 9 it is suggested that when the TAPEX model is initialized with a typhoon close to or having already made landfall that this affects the track accuracy and hence the rainfall accuracy. I understand that this could be problematic for forecasts between 1-24 hours lead time but would this issue not diminish with increasing lead time? For example the forecasts used in figure 8 (lead time 49-72 h), would these not have been initialized when the typhoons were yet to make landfall? My confusion here is linked to my confusion about the dtg in the previous point, so it would be greatly appreciated if the authors could provide clarification.

Response:

We appreciate for the Reviewer's comments. The authors addressed the issue that when the vortex was initialized near the complex terrain, the current technique we used in TAPEX might not perform as well as it does when the vortex was in the open ocean. This might introduce errors in the consequent precipitation forecast. Thus it explained our model's performance drop when the TAPEX model was initialized when a typhoon is close to or making landfall on Taiwan even if the forecast time is as small as 1 to 24 hours. The same issue may not create problems when the lead time is greater (or typhoon is away). However, other issues such as lack of observations cause the initial field degradation. The text is added in red accordingly in the revised manuscript. Please fine the text on page 9 from lines 12 to 21.

7. Provide a table of the skill scores calculated across all the typhoons This would give some useful overview statistics about the proposed system, it might also help the reader to determine how close the system is to providing a FAR<0.5 as alluded to by Coughlan de Perez et al., 2016 and at which threshold level.

Response:

We appreciate for the Reviewer's comments. Table 7 is added for Reviewer's reference in the revised manuscript. With the data assimilation technique, the forecast results of 1 to 24 h is applicable. When the possibility threshold increases, the FAR score decreases. It implies that the number of false alarms decreases. To meet Cough de Perez et al. (2016)'s standard, the system can provide FAR < 0.5 when

the possibility is above 30% with a lead time of 48 hours. For the lead time more than 48 hours, the system cannot meet the operational need of practitioner. A paragraph in red is added to address the finding at the end of the discussion section on page 11 from lines 21 to 27. The limitation of the system is also added in the same paragraph. As to the Reviewer's other minor typographical corrections, we have made the following modifications.

| Reviewer comment | Authors' response |
|---|---|
| 1. Page 1 line 10 replace 'preceding' with 'predicting'introduction part. In the abstract you should focus on your contribution. | Corrected |
| 2. Flood Awareness System (EFAS)' Page 3 line 6 the flash flood system in EFAS is currently the Enhanced Runoff Index based on Climatology (ERIC) see http://onlinelibrary.wiley.com/doi/10.1002/met.1469/abstract | Corrected |
| 3. Page 3 line 8 replace 'sources' with 'resources' | Corrected |
| 4. Page 4 line 8 replace 'early inundation warning system' with 'inundation early warning system' | Corrected |
| 5. Page 6 line 9 insert 'the' after 'The complexity of' | added |
| 6. Page 7 equation 3 replace 'Pobability' with 'Probability | Corrected |
| 7. Page 11 line 22 replace 'will appear' with 'are likely' | Corrected |
| 8. Figure 3 replace 'Present Time' with 'Timeline' | Corrected |
| 9. Table 1 caption insert 'that' so that the sentence reads 'For typhoons that did not...' | added |
| 10. Table 5, in Matmo SR-FAR column for the 80% exceedance level did SR-FAR really equal 1.0? | Yes, it equaled 1.0 since the system did not produce false alarms. There were misses, but not included in the performance measure. |

**Responses to reviewer #2**

In fact, the authors raised an essential point of research in their manuscript which is considered as (from my point of view) the highest priority for the hydrologists. I go through the paper several times in order to put my hand on how the authors introduced their contribution, I found that, the contribution is presented but not clear enough for me. Actually, while reading, there are several questions have been raised and also the difficulties in following the abbreviations reported in the text, so, first I recommend the authors to prepare a list of abbreviations at the beginning of the manuscript to easy following the manuscript. In addition, to avoid any duplication in the comments, I go through the first reviewer's report, I totally agreed with his comments and I am sure if the authors consider them, the manuscript will be in excellent shape for the readers.

Response:

We appreciate for the Reviewer's comments. We have revised the manuscript accordingly. A list of abbreviations was added after the Acknowledge section. Please see the attached manuscript for your reference. All the modification for Reviewer one's comment was made in red in the revised manuscript.

However, I am highlighting hereafter some other comments as well.

1. The formulation of the rainfall threshold is not clear, showing comprehensive details on how they form it is essential for the readers.

   Response:

   We appreciate for the Reviewer's comments. Figure 2 is added to provide further information on the identification process of WRA's rainfall thresholds. The text is also added in Section 2.2 in red. Please find the description on page 4 from lines 16 to 21 and lines 32 to 34 in the revised manuscript.

2. The introduction section is really very long, I suggest to shorten it to be direct to the point, to let the readers captured your idea in short way. Unless, the authors could split this section into several representative sub-sections.

   Response:

   We appreciate for the Reviewer's comments. The introduction section was shortened according to the Reviewer's comment. Please see the revised manuscript for your reference. The modification was made in red in the first section in the revised manuscript.

3. The title of section 4-1 is not understandable

   Response:

   We appreciate for the Reviewer's comments. The section 4-1 described the forecast results without a data assimilation technique which was proposed by the study. To be clarified, the title is changed to "Original forecast results without a data assimilation technique".

4. In section 4-2,,, at the beginning the authors start with the following

statement "To decrease the uncertainty of numerical weather predictions and improve the performance of inundation alert forecasting, this study developed a hybrid real-time observed and forecasted rainfall model to improve the accuracy of early warning notifications.". This section supposed to be results and discussion section, but I did not see before that how the authors develop this hybrid real-time observation model.

Response:

We appreciate for the Reviewer's comments. The "hybrid real-time observed and forecasted rainfall model" actually means that the system modified forecasts with a data assimilation technique. The method was explained from Line 25, Page 5 in the original manuscript. To be clarified, the authors modified the content. At Line 25, Page 5, the first sentence is also modified "To decrease the uncertainty in the rainfall forecasts, this study a data assimilation technique that used real-time rainfall observations to modify the rainfall forecasts and improve the 24-h urban inundation forecast's performance." At beginning of Section 4.2, the content is modified "To decrease the uncertainty of numerical weather predictions and improve the performance of inundation alert forecasting, this study applied a data assimilation technique that combined real-time observed and forecasted rainfalls to modify the forecasts. The data assimilation technique decreased the temporal uncertainty of numerical rainfall forecasts and improved the accuracy of early warning notifications. The longest rainfall threshold duration to trigger an inundation alerts is 24 h in this study. The technique was used to address the gap in forecasted rainfall data with observed rainfall information. The absence of forecasted rainfall values occurred in the first warning period (i.e., 1-24 h). Therefore, this study used the data assimilation technique to improve the 1- to 24-h forecasts."

5. Presentation of figures 5 and 6 are not of good quality. Figure 5 could be improved and increasing its scale. Figure 6, its notation is wrong (a, b) is repeated: : : and c the number 15 is appear which is not understandable for what.

Response:

We appreciate for the Reviewer's comments. The figures are modified

in the revised manuscript. The authors reproduced Figure 5 and corrected the notation in Figure 6.

6. Adding a paragraph at the end of discussion section showing the limitations of the proposed method would be very helpful for readers

   Response:

   We appreciate for the Reviewer's comments. The limitation of the system is added in red at the end of the discussion section. In addition, Table 7 is added based on Reviewer #1's request to address the overall discussion of the system. With the data assimilation technique, the forecast results of 1 to 24 h is applicable. When the possibility threshold increases, the FAR score decreases. It implies that the number of false alarms decreases. To meet Cough de Perez et al. (2016)'s standard, the system can provide FAR < 0.5 when the possibility is above 30% with a lead time of 48 hours. For the lead time more than 48 hours, the system cannot meet the operational need of practitioner. The discussion is added to address the finding at the end of the discussion section. Please find the modification on page 11 from lines 21 to 27 in the revised manuscript.

7. The conclusion section is also very long and include several parts that consider as a discussion issue and not consider as conclusion, better to re-write this section to be direct to the point and reflect the objective of the study.

   Response:

   We appreciate for the Reviewer's comments. The conclusion section is condensed to two paragraphs. The first paragraph describes the overall picture of the study and general findings from the original results without a data assimilation technique. The general recommendations of using numerical rainfall forecasts during typhoons were proposed at the end of the first paragraph. The second paragraph then describes the improvement of the system after considering real time observations in the evaluation process. Finally, some suggestions are included for the future improvement. Please find the modification which was made in red at the conclusion section in the revised manuscript.

[revised manuscript text omitted]